# Thioredoxin Reductase Activity Predicts Gold Nanoparticle Radiosensitization Effect

**DOI:** 10.3390/nano9020295

**Published:** 2019-02-19

**Authors:** Sébastien Penninckx, Anne-Catherine Heuskin, Carine Michiels, Stéphane Lucas

**Affiliations:** 1Research Center for the Physics of Matter and Radiation (PMR-LARN), Namur Research Institute for Life Sciences (NARILIS), University of Namur, Rue de Bruxelles 61, B-5000 Namur, Belgium; sebastien.penninckx@unamur.be (S.P.); anne-catherine.heuskin@unamur.be (A.-C.H.); stephane.lucas@unamur.be (S.L.); 2Unité de Recherche en Biologie Cellulaire (URBC), Namur Research Institute for Life Sciences (NARILIS), University of Namur, Rue de Bruxelles 61, B-5000 Namur, Belgium

**Keywords:** gold nanoparticles, radiosensitization, thioredoxin reductase, radiation, prognosis, biochemical mechanism

## Abstract

Gold nanoparticles (GNPs) have been shown to be effective contrast agents for imaging and emerge as powerful radiosensitizers, constituting a promising theranostic agent for cancer. Although the radiosensitization effect was initially attributed to a physical mechanism, an increasing number of studies challenge this mechanistic hypothesis and evidence the importance of oxidative stress in this process. This work evidences the central role played by thioredoxin reductase (TrxR) in the GNP-induced radiosensitization. A cell type-dependent reduction in TrxR activity was measured in five different cell lines incubated with GNPs leading to differences in cell response to X-ray irradiation. Correlation analyses demonstrated that GNP uptake and TrxR activity inhibition are associated to a GNP radiosensitization effect. Finally, Kaplan-Meier analyses suggested that high TrxR expression is correlated to low patient survival in four different types of cancer. Altogether, these results enable a better understanding of the GNP radiosensitization mechanism, which remains a mandatory step towards further use in clinic. Moreover, they highlight the potential application of this new treatment in a personalized medicine context.

## 1. Introduction

Over the past century, radiotherapy has emerged as the main treatment modality for cancer [1]. This powerful technique is based on the induction of lethal cellular damages caused by ionizing radiation delivered to tumors. Even if successful, this approach is still limited by dose distribution heterogeneity causing side effects to healthy tissues surrounding the tumor. In this way, the research on new strategies to achieve a better tumor targeting and enhance the biological effectiveness of radiation is growing [2,3]. Pushed by the development of nanotechnology, the scientific community takes advantage of nanoscale materials as sensitizers for therapeutic applications. It was suggested that the strong difference in energy absorption between high Z nanoparticles and water could be used to increase the local dose deposition in cells [4,5,6]. The proof-of-concept was demonstrated by Hainfeld et al. [7] who evidence that injections of 1.9 nm gold nanoparticles (GNPs) increased the survival of tumor-bearing mice in combination with 250 kVp X-rays compared to X-rays alone. Since this pioneering work, the development of new high-Z radiosensitizers (including silver [8,9], gadolinium [10,11,12], hafnium [13,14], platinum [15,16], gold [6,17,18] or bismuth [19,20] nanoparticles) has accelerated and many studies have shown their ability, when injected into the tumor, to amplify the X-ray radiation treatment efficacy. While evidencing this potential use as a radiosensitizer, the large variations in all these experimental settings revealed the high variability of GNP effects according to different physico-chemical parameters including GNP size, shape and coating agent. Zhang et al. [21] performed radiosensitization experiments using four distinct polyethylene glycol (PEG)-coated GNP (5 nm, 12 nm, 27 nm, 46 nm). Although they showed that all GNP sizes caused a decrease in cancer cell survival after irradiation, they reported a stronger effect using the 12 nm and 27 nm GNPs due to a more important tumor accumulation. Moreover, other groups demonstrated the influence of coating agent and GNP shape in the cell uptake process and so, in their involvement in the radiosensitization effect [22,23].

Despite the increasing amount of data regarding GNP-induced radiosensitization, it is still difficult to draw conclusions regarding this radiosensitization effect due to the diversity of parameters and conditions (nanoparticle size, cell lines, radiation source, administration route, …) used in literature [2]. This leads to important open questions regarding the mechanism(s) responsible for this effect, which remains a mandatory step towards the clinical use of metallic radiosensitizers. In this context, the present work aims at shedding light on non-physical mechanisms responsible for the GNP-induced radiosensitization based on preliminary results described in our previous study [24]. In the present work, we focused on the effect of homemade 10 nm amino-PEG functionalized GNPs in five different cell lines (A431 epidermoid carcinoma, A549 lung adenocarcinoma, MDA-MB-231 breast adenocarcinoma, PANC-1 pancreatic epithelioid carcinoma and T98G glioblastoma cell lines). We evidenced correlations between GNP uptake, residual thioredoxin reductase (TrxR) activity and radiosensitization effect.

## 2. Material and Methods

### 2.1. GNP Synthesis

10 nm amino-PEG functionalized GNPs were synthesized according to reference [24]. Briefly, HAuCl_4_ (Sigma Aldrich, Overijse, Belgium) and TA-PEG_550_-OCH_3_ (Biochempeg Scientific Inc., Watertown, MA, USA) were mixed at a 2000:1 Au: PEG molar ratio in deionized water and stirred at room temperature for 1 h. NaBH_4_ (Sigma Aldrich) was then added to the mixture under vigorous stirring and the solution was left stirring for 3 h. Then, TA-PEG_400_-NH_2_ (Biochempeg Scientific Inc., Watertown, MA, USA) was added to the solution for extra passivation. After 3 h of stirring, the colloidal suspension was purified using a membrane filtration device (Vivaspin, Millipore, Darmstadt, Germany).

GNPs were lyophilized with a freeze-drying system (Alpha 2-4 LD Plus; Analis, Rhisnes, Belgium) and stored at 4 °C for further use. In all experiments, cells were incubated with 50 µg of gold per mL of medium, which corresponds to 8.22 nM of GNPs.

### 2.2. Cell Culture

Human lung carcinoma A549 cells were grown in Eagle’s Minimum Essential Medium (MEM Glutamax; Gibco^®^ by Life Technologies, Merelbeke, Belgium) supplemented with 10% (*v*/*v*) fetal bovine serum (FBS; Gibco^®^ by Life Technologies). Epidermoid carcinoma A431 cells, mammary gland adenocarcinoma MDA-MB-231 cells, glioblastoma T98G cells and pancreas epithelioid carcinoma PANC-1 cells were grown in Dulbecco’s Modified Eagle’s medium (DMEM 4.5 g/L glucose; Gibco^®^ by Life Technologies) supplemented with 10% (*v*/*v*) FBS. All cell lines were incubated at 37 °C in a humidified atmosphere incubator containing 5% CO_2_.

### 2.3. GNP Uptake

Gold content quantification was performed by atomic absorption spectroscopy (AAS, AA-7000F from Shimadzu, Kyoto, Japan). After a 24 h incubation with GNPs, the cells were washed twice with PBS at 37 °C and then harvested using trypsin. Detached cells were then washed twice with culture medium by successive centrifugation. The actual number of cells in each sample was determined using a cell counter (Countess Automated Cell Counter, Invitrogen, Merelbeke, Belgium). After the third centrifugation, the medium was discarded, and the pellets were digested using 2 mL of aqua regia (37% HCl, 65% HNO_3_ Sigma-Aldrich) overnight. The gold content was quantified using an atomic absorption spectrophotometer (AA-7000F from Shimadzu, Kyoto, Japan) by plotting the calibration curve with known concentrations of a gold standard solution (Merck Chemicals, Overijse, Belgium) in aqua regia-solubilized cells used for external calibration. Triplicate readings were analyzed for each sample. The amount of gold detected in the cells was expressed as an gold quantity (pg) per cell. Using the theoretical mass of a 10 nm GNP (=1.01 × 10^−17^ g), results were expressed as a number of GNPs per cell.

### 2.4. X-ray Irradiation

48 h before irradiation, 5 × 10^4^ cells were seeded as 50 µL drops in 24-well plates and placed in an incubator at 37 °C with 5% CO_2_. 2 h after seeding, the wells were filled with corresponding medium and placed in the incubator overnight. The medium was then removed and the wells were filled with medium + 10% FBS without (control cells) or with 50 µg Au·mL^−1^ of GNPs and incubated at 37 °C until irradiation (24 h of incubation). Prior to irradiation, the medium was discarded from the wells, the plate was rinsed with PBS and filled with CO_2_-independent medium (Gibco^®^ by Life Technologies) without FBS. The cell monolayer was irradiated with a homogenous X-ray beam produced by an X-Rad 225 XL (PXi Precision X-ray, North Branford, CT, USA) at 225 kV. The dose rate was fixed to 3 Gy·min^−1^ and the dose to 2 Gy.

### 2.5. Clonogenic Assay

Immediately after irradiation, the cells were detached using 0.25% trypsin and counted. In order to obtain countable colony numbers, the cells were seeded in 6-well plates containing medium supplemented with 10% FBS, penicillin/streptomycin and incubated at 37 °C. In parallel, cells were also seeded in separate dishes. 2 h after seeding, they were fixed with 4% paraformaldehyde (Merck Chemicals) for 10 min and washed with PBS 3 times. The cells attached to the dish were counted manually under an optical microscope to obtain the precise number of cells seeded. Eleven days post-irradiation, the colonies were stained with violet crystal in 2% ethanol. The number of visible colonies (containing 50 or more cells) was considered to represent the surviving cells, which were counted manually. The plating efficiency (PE) was calculated by dividing the number of colonies by the initial numbers of seeded cells. The survival fraction was obtained as the PE ratio for irradiated cells to the PE for control cells. The control cells underwent the same procedure except the irradiation step. At least three independent experiments were performed and the errors were evaluated as standard error of mean (SEM). In order to quantify the GNP ability to enhance cell death, the amplification factor (AF) was calculated as previously described [6].

### 2.6. TrxR Activity Assay

The TrxR activity was measured with a commercially available kit (Sigma Aldrich). The assay is based on the catalytic reduction of 5,5′-dithiobis(2-nitrobenzoic) acid to 5-thio-2-nitrobenzoic acid by TrxR. This reduction generates a yellow colored product. Its absorbance is measurable at 412 nm by spectrophotometry. The cells were incubated 24h with or without 50 µg Au.mL^−1^ of GNPs before to being detached with 0.25% trypsin. The cells were pelleted by centrifugation (1000 rpm, 5 min, 4 °C) and the medium was discarded. The pellet was resuspended in a homemade lysis buffer (9% *w*/*w* sucrose; 5% *v*/*v* aprotinin (Sigma-Aldrich), in deionized water) and disrupted by a dounce homogenizer. Then, the TrxR activity was measured according to the manufacturer’s instructions. The linear increase in absorbance at 412 nm was measured during 10 min using a spectrophotometer (Ultrospec 8000; GE Healthcare, Chicago, IL, USA). The TrxR activity rate was calculated from the slope of absorbance at 412 nm versus time. Data are plotted as mean absorbance values normalized by the total protein in the sample.

### 2.7. Patient Survival Analysis

The online SurvExpress gene expression database (http://bioinformatica.mty.itesm.mx:8080/Biomatec/SurvivaX.jsp) [25] was used for the analysis of overall survival in different cancer types (1296 samples in total). Patients were classified into two risk groups according to their TXNRD1 gene expression. The median in gene expression was used as the cutoff. The association between TXNRD1 expression and overall patient survival was assessed by using the Kaplan-Meier method and the significance was analyzed using the log-rank test. *p* < 0.05 was considered to indicate a statistically significant difference.

### 2.8. Statistical Analysis

All experiments were repeated at least three times on separate days. A one-way analysis of variance (ANOVA) was performed using Origin 8 (OriginLab, Northampton, MA, USA) in order to compare the differences between groups. The number of asterisks in the figures indicates the level of statistical significance as follows: * *p* < 0.05, ** *p* < 0.01, *** *p* < 0.001.

## 3. Results

### 3.1. GNP Uptake

To assess the GNP uptake in each cell type after a 24 h incubation, AAS measurements were performed. As illustrated in Figure 1, a gold content of 0.51 ± 0.07, 0.71 ± 0.18, 0.84 ± 0.17, 0.97 ± 0.08 and 2.0 ± 0.4 pg Au/cell was measured for PANC-1, A431, MDA-MB-231, T98G and A549 cells respectively, revealing a cell type-dependent uptake. Moreover, no significant toxicity was observed in any studied cell lines (Appendix A).

### 3.2. GNPs Decrease the TrxR Activity in Different Cell Lines

The enzymatic activity of TrxR was evaluated in the different cell lines incubated with or without 50 µg·mL^−1^ of GNPs during 24 h. As shown in Figure 2, a decrease in TrxR activity was observed in all the cell lines when incubated with GNPs. Moreover, results demonstrated that the level of this enzymatic inhibition is cell type-dependent with a 49 ± 7%, 64 ± 5%, 75 ± 4% and 88 ± 7% of residual TrxR activity for A431, T98G, MDA-MB-231 and PANC-1 cells respectively. However, one-way ANOVA (Tukey test) evidenced that this inhibition was not significant for PANC-1 and MDA-MB-231 cells. It must be noted that a 28 ± 3% residual activity was previously measured in A549 cells incubated with the same GNPs [24].

### 3.3. GNPs Enhance the Cell Death upon Irradiation

The five cancer cell lines were pre-incubated during 24 h with or without 50 µg·mL^−1^ of GNPs prior to be irradiated using 225 kVp X-rays. Cell survival fractions were assessed by standard clonogenic assays. As shown in Figure 3, survival fraction decreased in all cell lines when they were pre-incubated with GNPs. To quantify this decrease in survival fraction, we calculated the amplification factor (AF) which indicates the enhanced proportion of dead cells in the presence of GNPs compared with irradiation alone for a given dose. At 2 Gy, a clinically relevant dose per fraction, a 13 ± 4%, 23 ± 1%, 7 ± 4%, 14 ± 3% and 2 ± 1% AF was calculated for, respectively, A431, A549, MDA-MB-231, T98G and PANC-1 cells.

### 3.4. Correlation between Cell Response to Radiation, TrxR Activity and GNP Uptake

To better understand the relationship between survival fractions, gold content in cells and the GNP-induced TrxR inhibition, correlation studies were performed. Results highlighted a strong correlation between GNP uptake and the amount of inhibited TrxR (Figure 4A, Pearson’s r = 0.991), between the AF at 2 Gy and the residual level of TrxR activity (Figure 4B, Pearson’s r = −0.978) and between the AF at 2 Gy and the GNP uptake (Figure 4C, Pearson’s r = 0.872).

### 3.5. TXNRD1 Is an Unfavorable Prognostic Factor

To investigate the possible involvement of TXNRD1 (gene coding for TrxR) expression in patient survival, we retrospectively analyzed microarray datasets of different types of cancer. A total of 1,296 samples with TXNRD1 status from five datasets were taken into account in this study. Five independent clinical cohort datasets were used: GSE-42669 [26] and GSE-30219 [27] cohorts for glioblastoma and lung adenocarcinoma respectively, TCGA-BRCA and TCGA-HNSC cohorts for invasive breast and squamous cell carcinomas respectively as well as PACA-AU for pancreatic ductal adenocarcinoma. The characteristics of these different cohorts are described in Table 1. Using SurvExpress tools, overall patient survival was analyzed according to TXNRD1 mRNA expression. Figure 5 shows the Kaplan-Meier curves for different cancer types while a box plot across the groups is shown in Appendix A. The results showed that high expression of TXNRD1 gene was significantly associated with poor overall patient survival in brain, breast, lung and head & neck. The effect was the most pronounced for lung adenocarcinoma, for which the median survival time decreased from 102 months (low TXNRD1 expression) to 44 months (high TXNRD1 expression).

## 4. Discussion

For the past several decades, oncology has been shifting towards a personalized medicine approach where the treatment selection for each cancer patient is becoming individualized [28]. This medical field is rapidly evolving driven by networks such as “The Cancer Genome Atlas” (TCGA) or “Gene Expression Omnibus” (GEO) that enable to catalogue genetic changes in key genes associated to cancer. These datasets permit the scientific community to identify new potential biomarkers and therapeutic targets [28,29]. In this study, Kaplan-Meier survival plots generated from cohort data demonstrated that higher expression of TXNRD1 gene is significantly correlated with poor survival outcome, identifying this gene as an unfavorable prognostic factor for cancer patients. The protein encoded by TXNRD1 belongs to the pyridine nucleotide-disulfide oxidoreductase family and is a member of the thioredoxin (Trx) system. This system is a major redox regulator and comprises an oxido-reductase enzyme (TrxR) which catalyzes the reduction of oxidized Trx by coupling with the oxidation of NADPH to NADP^+^ [30]. Since TrxR is also involved in tumor growth and DNA replication [31,32], it is not surprising that TXNRD1 overexpression has been evidenced in many aggressive tumors [30,33]. Moreover, breast cancer resistance to docetaxel has been demonstrated in tumor expressing high mRNA TrxR level [34,35].

In the context of personalized medicine, a therapeutic strategy to treat overexpressing TXNRD1 tumors could involve the use of Trx system inhibitors such as auranofin. Several studies have shown the ability of auranofin to trigger ROS overproduction and apoptosis in different cell lines [36,37,38] and to exert an antitumor activity in mice bearing breast or lung xenografts [38,39]. These works paved the way to ongoing lung cancer [40], leukemia [41] or ovarian cancer [42] clinical trials. Although auranofin was already FDA approved for the treatment of rheumatoid arthritis, its use is associated to cytotoxicity in vitro as demonstrated by Wang et al. (IC_50_ values of 19 and 11 μM for 4T1 and EMT6 cells respectively) [38]. One less toxic alternative could be GNPs. This study reported a cell type-dependent TrxR inhibition that may be explained by differences in cell capacity to internalize GNPs as well as in basal TrxR expression in each cell line. Indeed, we showed that even if we measured a similar gold content in A431 and MDA-MB-231 cells, we observed differences in TrxR inhibition (49% of residual TrxR activity level in A431 cells versus 75% in MDA-MB-231 cells). These differences may be due to a lower TrxR expression in A431 cells compared to MDA-MB-231 cells as suggested by activity measurement in Figure 2 (0.026 versus 0.066 A.U./min. µg of protein for A431 and MDA-MB-231 cells respectively).

Our work highlighted a cell-dependent radiosensitization effect with 225 kVp X-ray photons enabling to eradicate up to 23% more cells (in case of A549 cells) at 2 Gy compared to irradiation without GNPs. Although various works evidenced the GNP ability to enhance the biological effectiveness of radiation [6,24,43,44], the mechanism responsible for this effect remains poorly understood. Currently, two theories have been suggested. On one hand, a physico-chemical mechanism coming from the difference in energy absorption between gold and the surrounding soft tissues enables a dose enhancement in cells containing GNPs [5]. The interaction between the ionizing particles and high Z atoms can lead to the emission of low-energy electrons (LEE) from the nanoparticle [11,45] and the production of ROS [46,47]. This “ballistic” approach which predicts that the cell response would be directly correlated to the gold content, requires a direct interaction between the incident beam and the GNPs. However, a growing amount of simulation works evidenced that the number of hits in a cell containing GNPs is very low, especially in case of charged particles [48,49]. Consequently, the calculated physical enhancement effects of GNPs are very low compared to the radiosensitization effect observed in in vitro studies. Moreover, various studies have reported significant radiosensitization effect with megavoltage X-rays where little or no increase in overall dose deposition would be expected according to the theory [50,51,52]. This suggests that other mechanisms have greater contribution than physical interaction to the radiosensitization effect [48,49,53,54]. On the other hand, we previously demonstrated the involvement of the Trx system in GNP-induced radiosensitzation, suggesting a biological mechanism [24]. We performed an invalidation of the TrxR expression in A549 cells using siRNA technology leading to a residual 15% TrxR protein level. These invalidated A549 cells were irradiated without GNPs, evidencing a significant radiosensitization effect (AF of 30% at 2 Gy) [24]. Therefore, we suggested a new mechanism: following cell uptake through a receptor-mediated endocytosis, endosomes containing GNPs fuse with lysosomes. By decreasing pH inside the vesicle, lysosomes trigger a GNP degradation leading to the release of gold ions, well-known TrxR inhibitors. This inhibition induces various dysfunctions of pathways leading to a cytoplasmic ROS accumulation, a decrease in ATP production and DNA damage repair alterations. Irradiation of these weakened cells will cause DNA damage and an extra oxidative stress in cells with limited ATP stocks and detoxification systems [24]. Although this link between TrxR and GNP-induced radiosensitization was previously hypothesized to explain lung carcinoma cells radiosensitization, the present study validated it in four other cell lines. Indeed, correlation analyses (Figure 4) demonstrated that the radiosensitization effect is strongly correlated to the residual TrxR level. The aforementioned suggested mechanism is in agreement with an increasing number of works that have started to take nanoparticle impacts on cellular processes into account. By comparing a large amount of studies describing GNP-induced radiosensitization, Butterworth et al. [54] concluded that oxidative stress plays a central role in the radiosensitization effect. This hypothesis was confirmed by various groups which evidenced a reduction in radiosensitization effect when DMSO, a ROS scavenger was present upon irradiation [6,15,55,56]. Recently, protein disulphide isomerase, an enzyme catalyzing the formation and breakage of disulfide bonds in cysteine residues, was suspected to be a key mediator of the cellular response to GNPs [57]. Interestingly, some studies have showed that oxide nanoparticles have the ability to decrease DNA repair efficiency without any ROS production enhancement, on the opposite effect to metallic NPs [58,59]. This highlights the need to consider biological impacts of nanoparticles in further studies in order to rationalize reported differences in literature and to progress in our global understanding of the phenomenon.

It must be noted that all the results in this work are based on in vitro cell culture studies. Hence, this experimental set up displays some limitations. We evidenced that the TrxR inhibition increases with the gold uptake in the cells. However, the GNP delivery to tumors in vivo at similar concentrations is much more challenging. This will require tumor cell targeting strategies such as GNP surface modifications with antibodies [60]. Moreover, in vivo studies are required to investigate if similar metabolic changes occur in tumor than the ones described in cells, when GNPs are injected into mice. Lastly, irradiations of cell monolayers performed in this study may not accurately represent the 3-dimensional in vivo setting.

## 5. Conclusions

This work highlights the importance of nanoparticle - cell interactions to fully understand the radiosensitization mechanism. It evidences that the implication of TrxR, previously reported in GNP-induced lung cancer cell radiosensitization, seemingly confirmed in other cancer types since a good correlation between cell response to radiation and residual TrxR activity level is highlighted. Overall, this would suggest that GNPs play a radiosensitizer role by weakening detoxification system in addition to the radioenhancer role widely described in literature. To progress in the development of nanotechnology for oncology applications, the capacity of nanomaterials to inhibit the thiol-reductases protein family (such as TrxR) and their impact on antioxidants need to be assessed. A deep understanding of the mechanism responsible for this enhancement effect still remains a mandatory step towards the optimized clinical use of nanomaterials as radiosensitizers.

## Figures and Tables

**Figure 1 nanomaterials-09-00295-f001:**
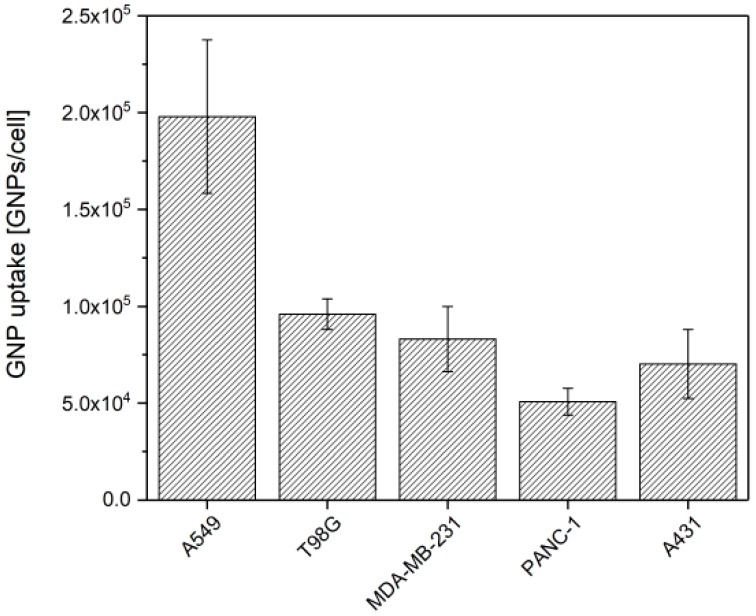
Cellular uptake of 10 nm GNPs in different cancer cell lines. Cells were incubated during 24 h with 50 µg·mL^−1^ of GNPs and the gold content was assessed by atomic absorption spectroscopy. Results are expressed as means ± SD for three independent experiments.

**Figure 2 nanomaterials-09-00295-f002:**
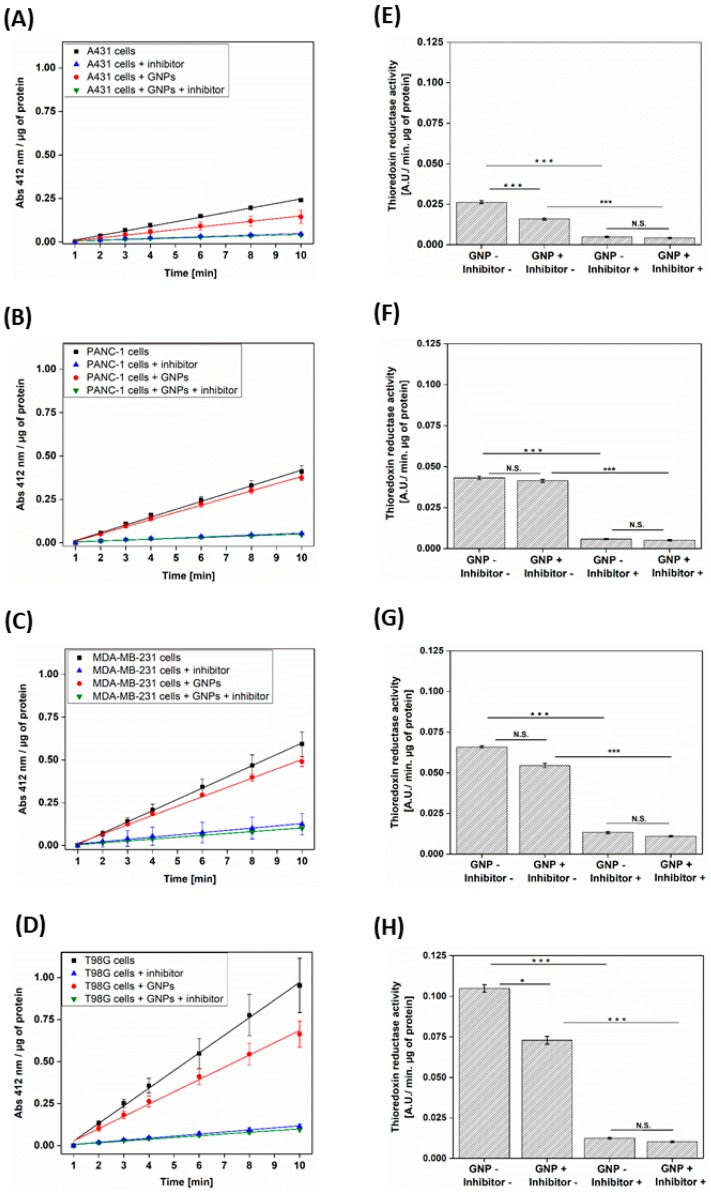
TrxR activity in cells incubated with or without 50 µg Au·mL^−1^ GNPs during 24 h. The activity was measured by the absorption at 412 nm over time in cell lysate of (**A**) A431, (**B**) PANC-1, (**C**) MDA-MB-231 and (**D**) T98G. Data are plotted as mean values of absorbance normalized by the total protein content ± S.D. of 3 independent experiments. Slopes of these TrxR activity curves were used to calculate the TrxR activity rate in (**E**) A431, (**F**) PANC-1, (**G**) MDA-MB-231 and (**H**) T98G cell lines. Data are plotted as mean values ± S.D. of 3 independent experiments. All results were statistically analyzed using a one-way ANOVA (Tukey test, * *p* < 0.05, *** *p* < 0.001, N.S. = not significant).

**Figure 3 nanomaterials-09-00295-f003:**
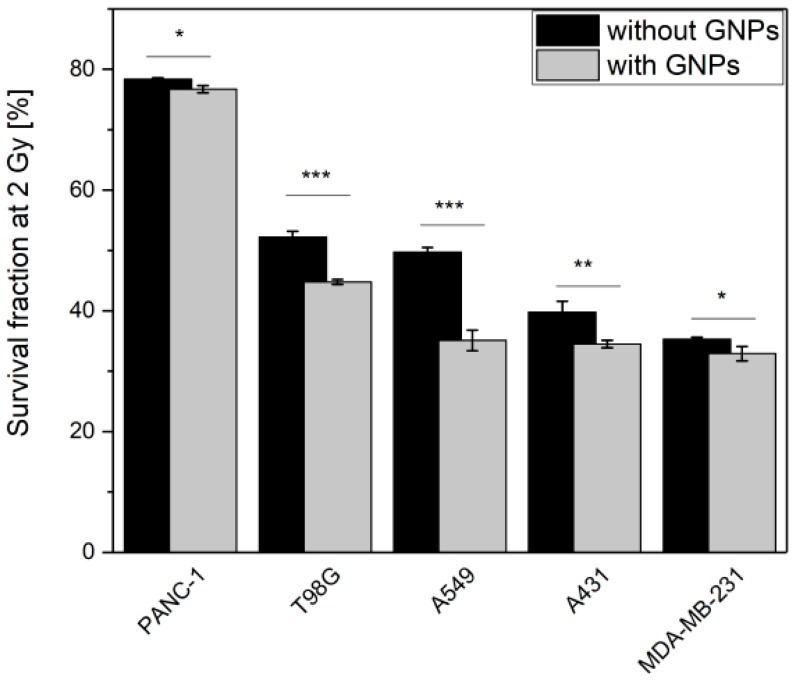
GNPs enhance cell death upon irradiation. Survival fractions were determined by colony forming assay for cells pre-incubated with 50 µg·mL^−1^ GNPs. After a 24 h incubation with GNPs, cells were irradiated with 225 kVp X-rays. Results are expressed as mean values of at least three independent experiments ± SEM. Data were statistically analyzed using a one-way ANOVA (Tukey test, * *p* < 0.05, ** *p* < 0.01; ** *p* < 0.001).

**Figure 4 nanomaterials-09-00295-f004:**
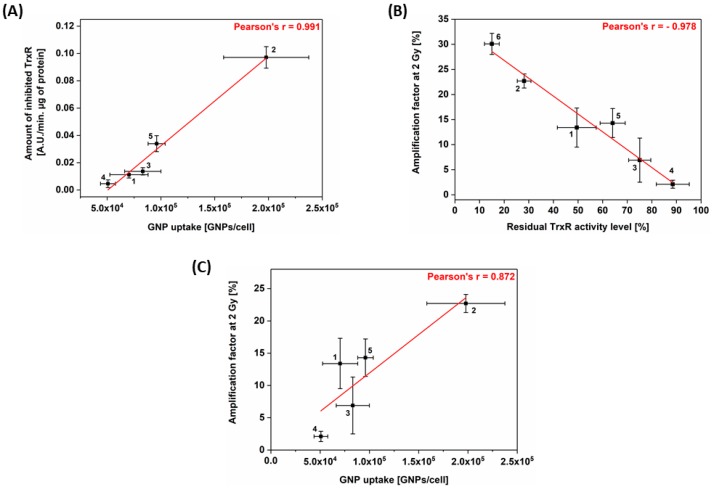
Correlation analysis showing the relation between (**A**) the amount of inhibited TrxR in cells and gold content; (**B**) the AF at 2 Gy and the residual TrxR activity level at irradiation time; (**C**) the AF at 2 Gy and the GNPs uptake. Data are presented as means of at least three independent experiments ± SEM (for AF) or ± SD (for TrxR activity and GNP uptake). 1 = A431 cells; 2 = A549 cells; 3 = MDA-MB-231 cells; 4 = PANC-1 cells; 5 = T98G cells; 6 = A549 cells invalidated for TrxR [24].

**Figure 5 nanomaterials-09-00295-f005:**
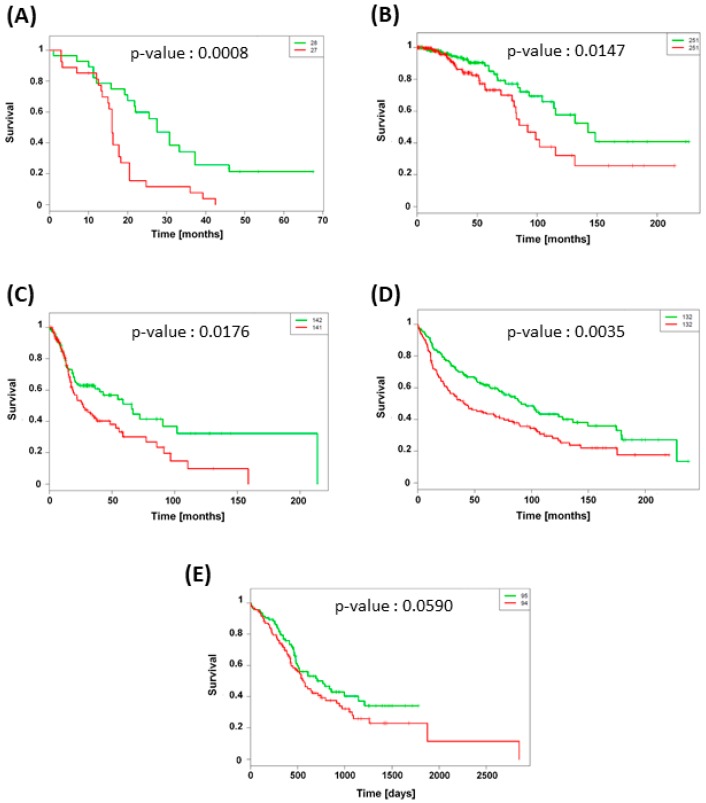
Kaplan-Meier analysis for overall patient survival according to TXRND1 mRNA expression in (**A**) brain (GSE42669), (**B**) breast (TCGA-BRCA), (**C**) head & neck (TCGA-HNSC), (**D**) lung (GSE 30219) and (**E**) pancreas (PACA-AU) cancer datasets. Green and red curves mean low and high TXNRD1 expression groups respectively. Log-rank equal curves p-values were calculated using the SurvExpress tools and were considered to be significant when *p* < 0.05.

**Table 1 nanomaterials-09-00295-t001:** Overall characteristics of the five datasets used in this study.

Organ	Cancer Type	Number of Patients	Median Survival Time (Months)		Database
			Low TXNRD1 Expression	High TXNRD1 Expression	
Brain	Glioblastoma	58	27	16	GSE 42669 [26]
Breast	Invasive carcinoma	502	142	92	TCGA-BRCA
Head & Neck	Squamous cell carcinoma	283	66	27	TCGA-HNSC
Lung	Adenocarcinoma	264	102	44	GSE 30219 [27]
Pancreas	Ductal adenocarcinoma	189	26	19	PACA-AU

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
