# Peer review of "Thioredoxin Reductase Activity Predicts Gold Nanoparticle Radiosensitization Effect"

_nanomaterials, 2019, doi:10.3390/nano9020295_

Round 1
Reviewer 1 Report
Several pieces of this puzzle have been presented, but not that well connected. In some cases, inappropriate comparisons have been made.
1) GNP uptake (line 89). Did the AAS gold calibration use the same matrix (addition of acid-solubilised cells)?
2) TrxR activity assay (line 125). Do GNPs affect the A412nm measurement, irrespective of TrxR activity?
3) GNP internalization (line 142). It is not possible to claim ‘internalization’ without TEM analysis. The authors should change this statement to 'in or on cells'. Furthermore, Figure 1 assumes that cells are the same size (volume), which may not be the case and therefore the number of GNP per cubic nanometre of cytoplasm may actually be similar across all the cell types tested.
4) TrxR activity (line 153). Goes back to my point about the activity assay. As a control, what happens to the A412nm values if GNPs are added to the already reacted samples? Does the presence of GNPs change the absorbance at 412nm independently?
5) Figure 2 (line 160). I'd prefer to see panels A-D and E-H with the same Y-axis scale, to facilitate comparisons between cell types.
6) Enhanced cell death with radiation (line 167). What about cell death in the absence of irradiation? Are there any purely chemotoxic effects of GNPs as measured by clonogenic assay? (Toxicity data in Supp 1 is only for 24hr and therefore can't be directly compared with clonogenic assays in Fig 3). Also, why does Supp 1 show ± 1 SD while other graphs show ±SEM?. I’d prefer Supp1 to be removed and for Fig 3 to include data on non-irradiated ±GNP cell death from clonogenic assays.
7) Also relating to irradiation (line 167). If TrxR activity is important for radiosensitisation, then this could be shown by irradiating ± TrxR inhibitors (such as Autothioglucose or Aurothiomalate). I know that this was done using a knockdown approach in another study by the authors, but what they want to do here is to compare the effectiveness of TrxR inhibition with irradiation across the different cell lines. For that, pharmacological inhibition would be perfectly suitable and straightforward to do.
8) Patient data (line 193). These data are difficult to interpret because we don't know what cancer treatment(s) the patients received. If it was possible to separate patients into irradiated and non-irradiated cohorts, then these data would be of more value. What was the criterion used to define 'high' or 'low' TXNRD1 expression?
9) Discussion (line 273). As mentioned in relation to line 197, it would be good to determine if a TrxR inhibitor (e.g. aurothioglucose) in the different cell lines replicates the radiosensitisation effect of GNPs. That would support the idea that GNPs are primarily acting as a delivery vehicle for gold salts that then inhibit TrxR; removing ROS protection and enhancing radiotoxicity.
Author Response
Manuscript ID: nanomaterials-447279
Thioredoxin reductase activity predicts gold nanoparticle radiosensitization effect
Sébastien PENNINCKX, Anne-Catherine HEUSKIN, Carine MICHIELS and Stéphane LUCAS
We would like to thank the reviewers for his/her helpful comments. We took into account all the comments in order to make the manuscript clearer, with a more precise interpretation of the results. We have also provided all the information requested by the reviewers. The changes performed are highlighted in the main text. An answer point by point to all the reviewers’ questions and comments are provided hereunder.
Reviewer 1:
1. GNP uptake (line 89). Did the AAS gold calibration use the same matrix (addition of acid-solubilised cells)?
The AAS gold calibration use the same matrix (addition of acid-solubilised cells) than the one used for the actual samples. We added this information in the material & method section of the revised version.
2. TrxR activity assay (line 125). Do GNPs affect the A412nm measurement, irrespective of TrxR activity?
The procedure for measuring TrxR activity involves a series of washing and centrifugation steps that enable to obtain a GNP-free final cell lysate. This is confirmed by the absence of plasmon band in cell lysate spectrum. However, as a control, we measured TrxR activity in two conditions:
- Cell lysate + TrxR inhibitor
- Cell lysate from cells incubated with GNPs + TrxR inhibitor.
As illustrated in figure 2, we did not observe a significant difference between these two conditions in all the cell lines tested. This result evidences that GNPs do not affect the TrxR activity measurement.
3. GNP internalization (line 142). It is not possible to claim ‘internalization’ without TEM analysis. The authors should change this statement to 'in or on cells'. Furthermore, Figure 1 assumes that cells are the same size (volume), which may not be the case and therefore the number of GNP per cubic nanometre of cytoplasm may actually be similar across all the cell types tested.
The word “internalization” has been modified in the revised version of the manuscript. The actual size of cells has been measured for all cell types used in this study by Z-stack in confocal microscopy. The cells used in this study have sizes comprised between 18 µm (MDA-MB-231) and 22 µm (PANC-1) leading to a number of GNPs/nm³ which is different depending on the cell type of interest. Moreover, we do not think that the number of GNPs/nm³ is relevant for the comparison since the distribution of GNPs inside the cells is not homogeneous: indeed, we observed a marked GNP clustering into vesicles, as evidenced by many other groups).
4. TrxR activity (line 153). Goes back to my point about the activity assay. As a control, what happens to the A412nm values if GNPs are added to the already reacted samples? Does the presence of GNPs change the absorbance at 412nm independently?
See answer to question 2.
5. Figure 2 (line 160). I'd prefer to see panels A-D and E-H with the same Y-axis scale, to facilitate comparisons between cell types.
The figure 2 has been modified according to the reviewer’s request.
6. Enhanced cell death with radiation (line 167). What about cell death in the absence of irradiation? Are there any purely chemotoxic effects of GNPs as measured by clonogenic assay? (Toxicity data in Supp 1 is only for 24hr and therefore can't be directly compared with clonogenic assays in Fig 3). Also, why does Supp 1 show ± 1 SD while other graphs show ±SEM?. I’d prefer Supp1 to be removed and for Fig 3 to include data on non-irradiated ±GNP cell death from clonogenic assays.
As suggested by the reviewer, figure S1 was replaced by toxicity data from clonogenic assay. No significant differences were reported for non-irradiated cells compared to non-irradiated cells with GNPs.
7. Also relating to irradiation (line 167). If TrxR activity is important for radiosensitisation, then this could be shown by irradiating ± TrxR inhibitors (such as Autothioglucose or Aurothiomalate). I know that this was done using a knockdown approach in another study by the authors, but what they want to do here is to compare the effectiveness of TrxR inhibition with irradiation across the different cell lines. For that, pharmacological inhibition would be perfectly suitable and straightforward to do.
8. Discussion (line 273). As mentioned in relation to line 197, it would be good to determine if a TrxR inhibitor (e.g. aurothioglucose) in the different cell lines replicates the radiosensitisation effect of GNPs. That would support the idea that GNPs are primarily acting as a delivery vehicle for gold salts that then inhibit TrxR; removing ROS protection and enhancing radiotoxicity.
We are grateful to the reviewer for this interesting comment. However, we think that an approach based on TrxR inhibitor is not the best suited one due to the large amount of off-target effects associated with these molecules. For example, auranofin inhibits TrxR but also the glutathione peroxidase, the glutathione-disulfide reductase, the nitric-oxide synthase, the phosphofructo-kinase and many other enzymes [1]. This large number of off-targets does not enable a relevant study. However, Wang et al. [2] investigated the radiosensitizing potential of auranofin in 4T1 and EMT6 cells. They pre-incubated tumor cells with auranofin at sub-cytotoxic concentration for 2 h before to expose them to various radiation doses. They reported an increased radiosensitivity with increasing auranofin concentrations.
Rather than using inhibitor, we chose a silencing approach using siRNA technology. This approach allows to specifically inhibit the expression of TrxR, hence to decrease its activity, without affecting the activity of the other enzymes. The invalidation of the TrxR expression was performed in A549 cells leading to a residual 15% TrxR protein level. These invalidated A549 cells were irradiated without GNPs, evidencing a significant radiosensitization effect (AF of 30% at 2 Gy) [3]. As illustrated on the figure below, this invalidation result (point 6) is in agreement with the correlation obtained with cells incubated in the presence of GNP (points 1 to 5). Adding this point even increases the Pearson’s r coefficient value (from -0.964 to -0.978).
Correlation analysis showing the relation between the amplification factor at 2 Gy and the residual TrxR activity level at irradiation time. 1 = A431 cells; 2 = A549 cells; 3 = MDA-MB-231 cells; 4 = PANC-1 cells; 5 = T98G cells; 6 = A549 cells invalidated for TrxR [3].
9. Patient data (line 193). These data are difficult to interpret because we don't know what cancer treatment(s) the patients received. If it was possible to separate patients into irradiated and non-irradiated cohorts, then these data would be of more value. What was the criterion used to define 'high' or 'low' TXNRD1 expression?
Although it is possible to separate patients according to their treatment type in some databases, this information is missing in the databases used in this study.
The median in TXNRD1 gene expression was used as cutoff to define ‘high’ and ‘low’ expression.
Reviewer 2 Report
The goal of this work was to shed light on the nonphysical mechanisms responsible for gold nanoparticle (GNP)-induced radiosensitization in 5 tumor cell lines. Enhancing the efficacy of external beam radiotherapy is an important aspect of investigation currently. Findings support the role of residual thioredoxin reductase activity, cellular GNP uptake, and subsequent radiosensitization of the cells to 2 Gy of external beam radiotherapy. The paper is well written and presented in a logical manner. The use of “evidenced” in numerous places appears out of context in many of the sentences and simple replacement with other more appropriate descriptors will be beneficial to readers. Patient survival cohorts appear appropriate for teasing out specific biologic mechanistic nuances and their impact on overall survival. Corroborating in vitro results with banked txnRD1 gene expression levels was creative. There were a few comments or concerns that would improve the paper in this reviewer’s opinion.
1. It would have been beneficial to include a normal cell line as a control along with the 5 tumor cell lines in the in vitro radiosensitization studies. The reader does not get a sense of the targeting accuracy of the GNP in vivo and of course it is not desirable to enhance the radiation sensitivity of normal cells during an in vivo radiation treatment.
2. The results are based on in vitro cell culture studies which are important in the big picture setting, but the authors need to clearly state that these results are based on in vitro findings and caution that getting the agent to tumors in vivo at similar concentrations and exposure times is much more challenging. This needs to be clearly stated in a study limitations paragraph.
3. Radiating cell monolayers externally may not accurately represent the 3-dimensional in vivo setting which is important given that a secondary radiation event is likely playing a main role in efficacy. By simple geometry, the majority of the peripheral secondary radiation dose would be lost to air above and below the cell monolayer plane. This may actually underestimate the effect. This should also be mentioned as a study limitation.
4. The ultra-detailed description of the potential biological effects of Gd on TrxR inhibition and subsequent impact on cell survival on the top of page 9 appears rather speculative at this point.
5. A conclusion is that “It evidences that the …….., is confirmed in other cancer types since a strong correlation between cell response to radiation and residual TrxR activity level is highlighted.” This may be overstated since the amplification factor in PANC-1 cells was only 2+1%. While the best result was 23%, amplification in the others was less than 14%. This conclusion needs to be toned down a bit.
Author Response
Manuscript ID: nanomaterials-447279
Thioredoxin reductase activity predicts gold nanoparticle radiosensitization effect
Sébastien PENNINCKX, Anne-Catherine HEUSKIN, Carine MICHIELS and Stéphane LUCAS
We would like to thank the reviewers for his/her helpful comments. We took into account all the comments in order to make the manuscript clearer, with a more precise interpretation of the results. We have also provided all the information requested by the reviewers. The changes performed are highlighted in the main text. An answer point by point to all the reviewers’ questions and comments are provided hereunder.
Reviewer 2:
1. It would have been beneficial to include a normal cell line as a control along with the 5 tumor cell lines in the in vitro radiosensitization studies. The reader does not get a sense of the targeting accuracy of the GNP in vivo and of course it is not desirable to enhance the radiation sensitivity of normal cells during an in vivo radiation treatment.
We are grateful to the reviewer for this interesting comment. In this work, we did not discuss the tumor cell targeting and potential effects of GNPs on normal cell to avoid digressing too much from the main focus of our study: the role of thioredoxin reductase in the radiosensitizing effect. However, we have investigated the two topics pointed out by the reviewer. Recently, we published an article on the issue of cancer cell targeting by grafting an EGFR antibody (Cetuximab) on gold nanoparticle surface [4]. We reported that the conjugated nano-objects (Ctxb-GNPs) specifically bound to and accumulated in EGFR-positive cells compared to EGFR-negative ones. Moreover, Ctxb-GNPs enhanced the effect of irradiation in EGFR-positive cells but not in EGFR-negative ones.
In the meantime, we evaluated the GNP effects on three normal cell lines and evidenced a cell-dependent thioredoxin reductase inhibition such as for cancer cell lines. These results are described in another manuscript which is still under review. It must be stated that it is difficult to develop actual clonogenic assay for most of the normal cell types, due to their limited proliferation capacity. Hence, it is not possible to directly evaluate the radiosensitization effects of NPs on normal cells.
2. The results are based on in vitro cell culture studies which are important in the big picture setting, but the authors need to clearly state that these results are based on in vitro findings and caution that getting the agent to tumors in vivo at similar concentrations and exposure times is much more challenging. This needs to be clearly stated in a study limitations paragraph.
As requested by the reviewer, a paragraph regarding the limitations of our study has been added in the revised version of the manuscript (pp.276-282).
3. Radiating cell monolayers externally may not accurately represent the 3-dimensional in vivo setting, which is important given that a secondary radiation event is likely playing a main role in efficacy. By simple geometry, the majority of the peripheral secondary radiation dose would be lost to air above and below the cell monolayer plane. This may actually underestimate the effect. This should also be mentioned as a study limitation.
As requested by the reviewer, a paragraph regarding the limitations of our study has been added in the revised version of the manuscript (pp.276-282).
4. The ultra-detailed description of the potential biological effects of Gd on TrxR inhibition and subsequent impact on cell survival on the top of page 9 appears rather speculative at this point.
We are sorry, but we do not understand this comment. Indeed, we did not discuss the potentiel biological effects of Gd in our manuscript.
5. A conclusion is that “It evidences that the …….., is confirmed in other cancer types since a strong correlation between cell response to radiation and residual TrxR activity level is highlighted.” This may be overstated since the amplification factor in PANC-1 cells was only 2+1%. While the best result was 23%, amplification in the others was less than 14%. This conclusion needs to be toned down a bit.
The conclusion was toned down as suggested by the reviewer.
In the light of the aforementioned responses, we think that the revised version of the manuscript is now worth to be published in Nanomaterials. This topic can be of high interest to transdisciplinary teams working on development of nanotechnologies associated to biomedical applications. We are convinced that Nanomaterials, as a pioneer in this topic, is the best journal suited to that purpose.
Thanking you in advance for considering this work,
Sincerely yours,
Mr Penninckx Sébastien
References
1. Purich, D., The Inhibitor Index: A Desk Reference on Enzyme Inhibitors, Receptor Antagonists, Drugs, Toxins, Poisons, Biologics, and Therapeutic Leads. 2017: CRC Press.
2. Wang, H., et al., Auranofin radiosensitizes tumor cells through targeting thioredoxin reductase and resulting overproduction of reactive oxygen species. Oncotarget, 2017. 8(22): p. 35728-35742.
3. Penninckx, S., et al., The role of thioredoxin reductase in gold nanoparticle radiosensitization effects. Nanomedicine (Lond), 2018.
4. Li, S., et al., Antibody-functionalized gold nanoparticles as tumor targeting radiosensitizers for proton therapy. Nanomedicine (Lond), 2019.
Round 2
Reviewer 1 Report
The authors have dealt with most of my queries and the manuscript is now suitable for publication.